# A Narrative Review of Haptic Technologies and Their Value for Training, Rehabilitation, and the Education of Persons with Special Needs

**DOI:** 10.3390/s24216946

**Published:** 2024-10-29

**Authors:** Eloy Irigoyen, Mikel Larrea, Manuel Graña

**Affiliations:** 1Systems Engineering and Automation Department, Bilbao School of Engineering, University of the Basque Country (UPV/EHU), 48013 Bilbao, Spain; eloy.irigoyen@ehu.eus; 2Group of Computational Intelligence, Faculty of Engineering of Gipuzkoa, University of the Basque Country (UPV/EHU), 20018 San Sebastian, Spain; m.larrea@ehu.eus; 3Faculty of Computer Science, University of the Basque Country (UPV/EHU), 20018 San Sebastian, Spain

**Keywords:** childhood special education, haptic technology, learning process, educational tools, pedagogical rehabilitation

## Abstract

Haptic technologies are increasingly valuable for human–computer interaction in its many flavors, including, of course, virtual reality systems, which are becoming very useful tools for education, training, and rehabilitation in many areas of medicine, engineering, and daily life. There is a broad spectrum of technologies and approaches that provide haptic stimuli, ranging from the well-known force feedback to subtile pseudo-haptics and visual haptics. Correspondingly, there is a broad spectrum of applications and system designs that include haptic technologies as a relevant component and interaction feature. Paramount is their use in training of medical procedures, but they appear in a plethora of systems deploying virtual reality applications. This narrative review covers the panorama of haptic devices and approaches and the most salient areas of application. Special emphasis is given to education of persons with special needs, aiming to foster the development of innovative systems and methods addressing the enhancement of the quality of life of this segment of the population.

## 1. Introduction

Early in the 21st century, there were already proposals for futuristic human learning systems deployed over virtual digital environments [1]. At the time, the designs for these environments were predominantly graphical and visual. These proposals have evolved recently [2] into virtual and augmented reality environments that contemplate diverse perceptual modalities, including haptic interactions and other less common modalities, such as eye tracking [3]. As illustrated in Figure 1, digital virtual environments are full of tools that enhance user immersion and decision-making processes, including haptic devices. However, the conversation on education needs is far from being circumscribed to the digital domain, as, for instance, the discussion about the attention given to the education of the voice for children with special needs has been raised as a serious concern by some researchers [4]. Improving educational processes is an intrinsically multidisciplinary effort involving pedagogy, engineering, psychology, art, medicine, and electronics [5,6]. In addition, people with special educational needs often require special interaction elements [7,8], due to underdeveloped sensory, behavioral, or cognitive capabilities. For instance, non-standard resources, such as a “quiet room”, may prove beneficial for children with maladaptive behaviors [9,10], while the use of self-monitoring devices/apps may enhance the on-task concentration of other students [9,10].

Haptic technologies are concerned with the intensity, duration, location, displacement, and composition of contact sensing, including the generation of semantic representations of tactile sensory data or, more generally, with sensations generated from the skin (dermic sensations). Although haptic interaction has not received as much attention as visual interaction, its importance has emerged in recent times. In fact, it has been proven that the combination of visual and haptic interfaces enhances training for the development of visuo-motor skills [22] and the spatial coding of objects [23]. Other researchers have proposed haptic sensations as precursors of application-specific information, highlighting the importance of the sense of touch in library digital searches [24] or in gamified educational products intended to foster sustainable behaviors [25]. Furthermore, it has been shown that users can easily interpret multi-channel tactile signals [26]. The positive impact of haptic interaction in standard STEM curricula learning has also been recognized in recent reviews [27,28]. Even in the process of learning gymnastics tasks, users can benefit from haptic augmentation of verbal information [29]. Finally, haptics have been discussed as a potential tool for enhanced communication with blind–deaf persons [30].

Actually, haptic technologies have been part of virtual reality applications for some time, providing a powerful tool to enhance user immersion. Conversely, the intention and degree of engagement of a person handling a haptic device can be induced from the pressure, grip, and vibration of the user contact with the haptic device. Such contact quality evaluations improve the algorithmic interpretation of the haptic interaction in some applications. For example, it is possible to assess the degree of certainty with which an object is selected by the user in a virtual environment.

Haptics play a large role in the increasing trend to live in virtual worlds as proposed by large social media corporations [31]. But they are also a fundamental component of the virtual learning spaces and environments that are becoming standard tools in many areas of medicine and education in general [22,32,33]. In this paper, we provide a narrative review of the broad area of haptic technologies and applications, with a special emphasis on their use in the education of persons at risk of exclusion and with special needs. A strong motivation for this review was to foster the development of this kind of system for special education needs. An innovation ecosystem is lacking, even for people that should be evident targets for the development of innovative haptics-based systems, such as blind persons. Therefore, there is still much room for new ideas and applications.

## 2. Haptic Technologies

Figure 2 illustrates some of the disciplines that are closely related to developments in the field of haptic technology, either supporting the development of haptic devices or having some synergy with haptic devices that enhances the overall properties of the system, usually improving immersion in virtual scenarios. Actually, this universe of computational digital interaction tools and services is intimately entangled with developments in artificial intelligence and computational intelligence [34,35].

### 2.1. Artificial Skin

Human skin provides information about its environment, in addition to tactile information. It measures temperature, vibration, pressure (strength), and the intrinsic properties of object surfaces, such as texture, roughness, and area of contact. For instance, in order to increase immersion, in a recent haptic solution [13] a chemically induced skin-heating system was used, to give the user the impression of increasing temperatures in a virtual reality environment. Conversely, measuring the temperature of the human skin allows for evaluating how active and confident a person is while performing some kind of task.

There has been some recent activity in the design of artificial skins with abilities emulating the human skin, such as the developments done at the Ecole Polytechnique Federale de Laussanne (EPFL) [12] based on a soft artificial skin that provides haptic feedback and can instantly adapt to the user’s movements. The Tesla suit can provide haptic feedback by transcutaneous electrical nerve stimulation, which has been tested for correcting errors in full-body motion exercises [36], while Meta Reality Labs [31] are exploring new configurations of haptic gloves to provide sense of touch, including sensations related to texture, weight, and stiffness. Other instances of current approaches include (a) the Johns Hopkins University [11] electronic skin, which can restore sense of touch through the fingertips of prosthetic hands, (b) a light-based contact detection [37], which offers an imaginative alternative solution to the problem of integrating touch devices into robotic hands, and (c) another novel wearable device [17] suitable for being used to investigate perception in interactive tasks on individuals with and without sensory disabilities, among others. Other devices can be categorized as pseudo-skins that are physically close to the human skin and can generate haptic effects, such as wearable systems with neuromuscular electrical stimulation (NMES), which offer the potential to reproduce the physiological stress of actual exercises by working on the antagonist muscles [38]. Finally, recent research on the yarns that can be controlled to become haptic actuators, called haptiyarn by the authors [39], has opened up a broad spectrum of potential ways to transform the fabrics of everyday clothing into haptic devices that are in close contact with the skin and can be connected to virtual reality environments for enhanced immersion.

### 2.2. Pseudo-Haptics, Multimodal, and Visual Haptics

The integration of haptics interfaces with other sensory modalities has been extensively explored in human–computer interaction systems.

**Pseudo-haptics** induce haptic impressions from other sensory inputs, such as sound or images, or combinations of them. They have been explored as a means to respond to the interaction needs of populations that will be secluded due to current and coming pandemics [40]. They will be providing affordable means by which to achieve some skill training requiring tactile sensing in virtual learning environments. They have also been proven to enhance the user experience for mid-air text entry [41], and walking-in-place [42,43] in virtual reality environments. Early approaches included the generation of pseudo-haptic sticky sensation by modifying the mouse control-display gain when approaching a target on the screen [44] combined with vibratory stimulation.

**Multi-modal systems** combine various sensors, including haptics, to create cognitive synergies that enhance the immersive experience. For instance, the combination of haptic force feedback with visual deformation effects has been applied to improved identification and examination of virtual soft tissues [45,46], with applications in medical training. Similarly, the vibration of the mouse pointer on the screen has been used to enhance the vibro-tactile perception of rough surfaces [47]. Recent studies have shown that applying vibrations to the wrist and elbow tendons may enhance pseudo-haptic perception [48]. However, some studies have found age and gender differences in the effectiveness of pseudo-haptic tricks [49].

Interleaving sound effects with the haptic experience has some desirable effects, such as the improved perception of self motion on a treadmill [50] by the generation of simulated footstep sounds. The observer integration of audio and haptic perception of rhythm allows improved immersion combining both sensory inputs [51]. An innovative boozer presented in [16] can be used, for instance, to elicit small stimuli at low frequencies (up to 150 hz), in order to create a signal activation in a very specific context, either to induce attention or to stop a hyperstimulation condition suffered by a student under stress. In a similar approach, a device called Sensory Substitution [20] is based on a non-invasive technique to convey information about the environment to deaf people by means of a waistcoat that provides the user with this information through small vibrations applied on his/her skin. Alternatively, haptic sensations can be decoded into sounds [52] with great potential for application in games and communication with hearing-impaired persons [53].

Artificial vision techniques are the source of another kind of multimodal system that can significantly enhance haptic solutions by providing complementary information on the environment, including other subjects interacting with the user in virtual environments. For instance, the guidance of people with visual impairments can be carried out with an innovative mobility assistance system, which provides assistance to two main functions of navigation: locomotion and wayfinding [21]. At a biological level, there are proposals for sensory systems collecting optical and pressure information from both a photodetector and a pressure sensor, then transmitting the bimodal information through an ion wire and integrating it into postsynaptic currents that directly feed to the brain [18].

Other works [14] have integrated smell and haptics in medical training virtual systems, highlighting the important role of the sense of smell during both diagnosis and surgery in a medical setting. In artistic fields, recent studies have presented haptic technology as a step towards intensifying the virtual experience of viewing works of art. Another work, ref. [15], has presented mid-air haptics as a unique and first-time case study for engaging all senses (i.e., smell, sound, touch, vision, and taste) into the design of artistic experiences.

**Visual haptics** consists in the induction of haptic sensations through the perception of visual effects alone [54]. For instance, mimicking the visual dynamics of an avatar provides a surrogate of haptic sensation in a weight lifting task [55]. The apparent different velocity in the visual display in response to the haptic control is also used to induce sensations of friction and stiffness [56] in teleoperated surgery, and in virtual reality environments [57]. The cross-modal integration allows, for example, experience of the tactile deformation of an object as the user moves the hands, with the object showing greater or smaller deformation according to the distance between the hands [58]. Modifying the visualization of the dynamics of the objects that are virtually manipulated provides sensations about the mass of the objects, i.e., making more heavy objects rotate more slowly [59] or making the cursor response to the keystroke vary in different areas of the visualization [60]. The range of values of the ratio of the object movements to the user actions has been found to have an effect on the user’s mass perception [61]. Recent approaches in the area of virtual reality have tried to induce kinesthetic feedback sensation by the addition of an appendage to the avatar’s hand that translates object resistance into appendage deformations [62], and to apply deep learning for the estimation of manipulation forces from laparoscopic surgical images that are visualized, to provide visual haptic feedback to the remote user [63].

### 2.3. Alternative Tactile Sensations

Innovative haptic devices are capable of simulating a wide variety of tactile perceptions, which can be exploited for immersive virtual gaming experiences. For instance, delivering stimulus-generating liquids to the user’s skin creates different sensations [13]. In fact, the study has identified five chemicals that can produce specific dermic sensations: tingling (sanshool), numbing (lidocaine), stinging (cinnamaldehyde), warming (capsaicin), and cooling (menthol). Another approach to producing programmable thermal sensations (cooling and heating) is that of flexible adhesives that include layers of cooling hydrogel, electric resistances, and communications and control layers [64] so the system can be wirelessly connected to external controllers. A snake effect, i.e., inducing a sensation that can be interpreted as a snake moving over the skin, can be achieved by an array of tactors (motion effectors deployed over the skin) following specific vibrotactile spatio-temporal patterns [65]. Such vibrotactile sensations can be generated also by magnets embedded in soft tissue [66]. Contactless pilo-erection can be achieved by electrostatic devices inducing some haptic stimuli related to specific emotions, such as fear [67]. Although invasive solutions were not the focus of this review, we can refer to a remote touch system created to reproduce tactile stimuli in the user by means of electrical signals induced in the nervous system with special magnetic synapses [68] as a very special case of induction of dermal and tactile sensations.

### 2.4. Touchscreen Haptics

Interaction with the touchscreen where diverse patterns of tapping and dragging objects reveal diverse degrees of maturity in children [69] can be considered a specific kind of haptic experience that is very easy to measure for scientific purposes. Tactile stimulation (i.e., vibration of a smartphone) can be used, for instance, for covert interaction with subjects of special education, reducing embarrassing public interactions [70].

### 2.5. Interrelation of Body Motion and Haptics

Even gestures that are equivalent to virtual subjective haptic representation of the manipulation of the objects appear to have some influence on knowledge about the object [71]. This relation between vision, gesture, and haptics has been demonstrated in the training for the reinterpretation of visually ambiguous images [72,73]. The image mental model seems to be influenced by the motor models constructed by gestures and/or haptic interface interaction. The relation between language and motor models of the objects has been found to evolve with age [74], providing new clues about the evolution of fragility and some guidance on the use of haptic devices to assist in healthy aging research. Also, in regard to the early ages from 16 to 18 months, studies have been conducted on the correlation between visual and haptic responses that found indication of the evolution of world knowledge representation [75]. Over a greater span of ages, from 3 to 85 years, a brief assessment has found increasing accuracy in body mental representation from childhood to young ages, a slight decrease from young to older adults, and significant differences among adults and old adults [76].

## 3. Haptic Technology General Application Areas

Here, we provide a narrative revision of some of the hot areas of application of haptic technologies. Most of these areas are related to the use of virtual scenarios for learning, training, or rehabilitation purposes. The use of these techniques for education of people with special needs is the subject of another section.

### 3.1. Medicine

Haptic devices for medical skills training are being developed in many areas of medicine, as illustrated in Figure 3. This goes up to the proposal of a new kind of medical approach, so-called haptic medicine [77], where touch can be part of curative treatments. Developed wearable solutions enhance user perception processes for people with some kind of disability, like wearable devices composed of small units embedded with actuators that induce stimuli (haptic, light, and sound) to the users, and sensors that record the users’ responses [17]. On the other hand, haptic feedback is a highly valuable feature in some medical instruments, such as laparoscopic graspers [78]. Haptics have long been proposed for medical training, with some studies showing the influence of haptic feedback and the synchronisation of rendered haptics to improve user learning capability [79]. The positive effect of the inclusion of haptics in virtual simulation surgical education is increasingly recognized [80]. Dentistry is an area of medicine that has extensively exploited virtual training systems endowed with haptic features [32,81,82,83]. Nowadays, these kinds of systems have even entered the category of validated tools for virtual examinations, such as for dental technicians [84]. The increasing quality of haptics interfaces is very important for the introduction of virtual training systems in other areas of medicine, such as urology, where they have been proposed [33] to replace the cadaverical models.

In the same line of research, Deakin University presented a groundbreaking development, integrated into its HeroSurg robot [86], to give surgeons the sense of touch while a robot is driven via a computer to perform minimally invasive surgery.

Haptics have been included in multimodal virtual reality (MVR) systems for medical training, for instance in obstetrics, where these systems allow haptic expert knowledge to be transferred to the student [87]. Haptic feedback is also a fundamental element of neurosurgery training simulations [88,89,90], which are increasingly proposed as a mean for frontierless collaboration in the evolution of innovative procedures and techniques in neurosurgery [91].

The value of haptic feedback has also been widely recognized in endoscopic training, specifically endoscopic neurosurgery [92] and laparoscopic surgery [93,94,95], allowing automatic evaluation of the performance of the trainee [96]. Moreover, haptic microlaryngoscopy simulation training has also been positively evaluated by trainees [97]. Some very specific surgeries and medical procedures, like the retropubic midurethral sling [98], tympanomastoidectomy [99], or lumbar puncture [100], can be advantageously taught using haptically enhanced virtual reality processes. Haptics also plays a big role in teaching dental anesthesia in the inferior alveolar nerve block on a 3D printed model of the region [101,102], bone drilling in virtual reality training scenarios [103], or middle-cranial fossa dissection [104]. Furthermore, fusion of high resolution (eight Teslas) magnetic resonance imaging data with haptics has allowed high-resolution bone simulation in dissection for training of otological surgeons [105].

### 3.2. Games, Gamification, and Remote Control

New game interaction proposals from immersive virtual reality laboratories, such as the Meta touch solution [31] illustrated in Figure 4a, have become perfect tools to interact with the user and enhance sensory processes, opening new frontiers on developing advanced solutions in fields such as special education. The Teslasuit [85] is another example of the advancement of haptic technology in some areas, offering a wide range of solutions, such as suits and gloves. Nowadays, in a more general framework, it is possible to find communities in the haptic world where new proposals and ideas are aired, as in the case of Hapticast [106], as illustrated in Figure 4c. Current hot efforts in the development of haptics for games are mostly devoted to producing sensations of impacts on the body, such as bullets in warring games, or emotion-induction effects, such as the heartbeat during stressful situations within the game. However, these tricks are not of much use in games that have educational applications.

Haptic technologies play a substantial role in the gamification of educational contents. For example, Teenage Engineering has developed a Rumble subwoofer for one of the newer synthesizers, the OP-Z [16]. This haptic sound device can be inserted into a pocket, thanks to its miniaturization (Figure 4b). Another interesting experience is the one developed by the researchers and professors who collaborate in the Tec Monterrey CyberLearning & DataSciences Laboratory, located on the Mexico City Campus, where, through multimodal tactile interaction, they have shown how to improve the conventional teaching–learning processes [107].

Haptic sensation in games can be generated from sources, such as sound or telemetry data in car racing games [108]. Specifically, the use of telemetry data allows for simulating car crashes, the texture of the road, and the response of the car engine to the user commands.

A recent work [109] has shown that it is possible to increase the user’s awareness of an enveloping protection system in remote/virtual flight control where the user can obtain five signals: firstly, a discrete force signal when approaching the limits of use; secondly, an increased spring coefficient for control deviations at positions closer to the limits; thirdly, a shaking action for low speeds; fourthly, a movement adapted to the desired control input; and, finally, automatic operation when the limit conditions are exceeded.

### 3.3. Rehabilitation

In rehabilitation, haptic devices help patients to improve their motor and sensory apparatus, either through assistance with movement realization or by enhancing their environmental perception. Products like e-dermis [11] and SPA-skin [12] have very useful features that allow for control of the induction of some sensations, such as pain, pressure, temperature, etc., which are transmitted to the brain through peripheral nerves. It has been shown that motivation to engage positively in hard training, such as is needed for rehabilitation after a stroke, can be enhanced through social interaction via haptic interfaces [110]; however, the studies about the usefulness of robot-mediated haptic dyads—i.e., human–robot–robot–human systems—for motor rehabilitation in neurorehabilitation are inconsistent, reporting widely varying results [111]. Additionally, some experiences have shown a positive effect of the use of force feedback interfaces in neural rehabilitation processes [112].

### 3.4. Personal Assistance

Haptic technologies also play a big role in the development of assistive technologies for people with disabilities during the realization of their daily tasks. There are in the market some non-intrusive solutions that improve the lives of people with disabilities, such as WeWalk for the visually impaired, which vibrates to inform the user of low obstacles often missed by the bottom of a cane [19], or sensory substitution, which circumvents the loss of one sense by feeding its information through another sensory pathway [20]. On top of that, there are other more invasive solutions, such as generating haptic sensations by direct nerve excitation [68] via magnetic synapses, or the e-dermis proposal [11], which uses transcutaneous electrical nerve stimulation. Recent proposals for domestic robots to assist people with declining abilities have included haptics in their multimodal interaction interfaces as a way to achieve quick communication in extreme circumstances [113].

## 4. Haptics in Special Education Needs

The diversity of needs for education resources and tools is an ever-growing concern, due to the increasing social awareness of the diverse conditions of students at all ages. Technological solutions are increasingly required, to address such situations. For example, human–robot interaction opens up new avenues for educational interactions, where simple games like joint clapping of hands [114] can have a therapeutic effect on children with developmental syndromes. Conventional approaches often are out of the question in many cases and circumstances. For instance, in circumstances of pandemic confinement, face-to-face teaching is not allowed. Haptic interfaces have shown their value not only in these extreme circumstances but also in more common life situations where people have very stringent needs for which haptic communication has demonstrated specific value. Often, sensory deprivation opens up new and seemingly unrelated channels of communication. A recent review [115] showed that children with hearing impairment can also benefit their psychomotor development by the use of computer aiding systems including haptic interfaces. As another piece of evidence, it has been shown that the fusion of haptic information and movement has a principal effect in the embodiment sensation in virtual reality for children [116].

### 4.1. Visually Impaired People

For blind persons, tactile information is paramount for communication. For instance, variations of intelligence tests for visually impaired children were developed, based on early haptic interfaces [117], which were soon after proposed as a way to experience virtual reality scenarios by blind people [118]. Specifically, reading Braille displays is a required skill for blind people. Diverse tactile sensitivities and haptic feedback can be accommodated by adaptive Braille displays with adjustable cell sizes [119], and multi-array Braille displays combined with haptics can improve learning physical models [120]. Non-verbal communication information, such as facial emotional expressions, can be translated into vibrotactile information by artificial vision and conveyed to the visually impaired person [121]. Regarding spatial orientation, pin-array matrices allow compact representation of maps that can be tailored to enhance self-localization [122].

Even for deaf–blind persons, new haptic solutions are being proposed to enhance their communication channels. A glove endowed with vibrotactile modules can be activated by automatic speech-recognition systems, allowing close-to-real-time translation of speech into haptic signals [123]. For deaf–blind children, a haptic assistant allows more independent horseback-riding therapy [124].

Touch–vision interaction can work both ways. For instance, for children with visual impairment, their haptic training can be enhanced by their limited vision capabilities [125]. On the other hand, it has been found that children with visual impairment have a deficit in the motor representation of actions and objects [126], which can be treated with haptic assistants. A repertoire of haptic-based applications for middle-school students with visual impairment was recently developed and introduced experimentally in the classroom [127], to assist in the study of scientific and mathematical topics. For blind children, recent studies have shown the value of haptic virtual reality for teaching complex abstract concepts, such as 3D shape geometry [128].

### 4.2. Writing Aids

A critical motor skill that cannot be underestimated is writing. Haptic feedback has proved to be helpful in the acquisition and improvement of hand-writing skills [129] for children in early grades. Concerns for the development of this kind of skill have grown in recent years, due to the pervasive presence of screens, which has been proven to have a negative influence on the performance of mental imagery [130], and so new gaming applications have been proposed, to engage the children in handwriting activities [131].

Unfortunately, the current design of educational apps for tablets and mobile phones does not encourage writing skills and their corresponding visuo-motor development [132]. On the other hand, open source haptic devices for educational purposes that can reverse these trends have been proposed in recent years [133].

### 4.3. Motor-Impaired People

Motor deficits have negative impacts on academic performance at various levels. For instance, children with developmental coordination disorders have more difficulties in simple tasks, such as to appreciating the size of rods by pure haptic sensation [134]. As haptic perception is altered in cases of children with developmental coordination disorder [135] and children with developmental language disorder [136], the possible improvements that can be achieved by robotic assisted haptic training are of high value. Incidentally, table tennis has been found to be very helpful for such children [137].

Computer-aided training to perform the repetitive tasks required for handwriting learning has evolved to the extent that robotic systems are being proposed for routine skill improvement in children with motor difficulties [138] and for special education in general [131], due to the fact that they allow the tailoring of the training process to the idiosyncrasy of the child. Diverse kinds of haptic assistance (full haptic guidance, partial haptic guidance, disturbance haptic guidance) can be applied and have been found to be specifically useful for different tasks [139]. Also, training them in compliance with 3D haptic tracing tasks improved the 2D drawing abilities of children with motor difficulties [140]. Children with hemiplegic cerebral palsy have been stimulated to carry out rehabilitation exercises by immersive virtual reality games, including haptic feedback [141], achieving better engagement and partial recovery in several of the studies.

### 4.4. Cognitive Impaired People

Interest in haptic instrumentation for the educational rehabilitation of children with special needs emerged at the beginning of the computer era, which witnessed specific proposals for the treatment of children with non-typical mental development [142], following the early recognition of the role of active and passive touching in the process of building up perception of real objects [143], even for toddlers [144]. The importance of sense of touch in overcoming the limitations of a priori knowledge has become evident, such as was shown in an experiment where children were recognizing familiar and unfamiliar objects by haptic means [145].

The use of virtual reality techniques, specifically using haptic devices, has been considered for some time for several conditions, such as autism spectrum disorder (ASD), attention deficit hyperactivity disorder (ADHD), and cerebral palsy [146]. They can be applied as diagnostic assistant tools for some conditions, such as ADHD; however, the research on their use as assistants for treating the condition is still in its infancy. For example, a study of how ASD children regard agency and reward has been carried out using games on tactile platforms [147]. From a commercial point of view, the market has been flooded by a plethora of apps targeting ASD children, though most of them are of little value [148], even when the developers claim the use of artificial intelligence techniques.

However, haptics can play a major role in addressing these developmental syndromes. In this regard, haptic devices have been shown to have no detrimental effect in helping children diagnosed with ASD to transition between occupational therapy tasks [149], which can be shared with neurotypical development children. Haptic modeling of objects has been found to be more accurate in a small cohort of ASD children [150] that followed strategies similar to those of neurotypical children. A similar result was found in haptic-to-visual delayed shape matching in a cohort of adult ASDs [151], contrary to expectations. Additionally, higher-functioning ASD adults have been found to violate a central expectation in behavioral sciences, the so-called Weber’s law, in several perception tasks, including haptic weight discrimination, pointing to specific diagnostic/treatment tools in the future [152], such as improved quantitative measurement of hand–eye behavior for the evaluation of interactive tasks [153].

However, in daily contact with children we must be aware and respectful of their specificities. Technology is not the only solution, maybe not even the better solution. It has been shown that the use of quiet rooms seems to be highly beneficial for ASD children [9], alleviating the stress caused by the sensory-integration deficits suffered by ASD children. This effect, obviously, cannot be achieved with any technological solution that risks adding to the cognitive and sensorial saturation of the child.

## 5. Discussion

Haptic information can flow in two directions, to and from the user, in its interaction with artificial automated systems, which can be computers, robots, or other appliances, because haptics and its surrogates are involved in a plethora of interactive system designs. Most of the systems considered in the vast literature only consider one of these directions. For instance, new developments of haptic suits for gaming applications are focused on giving the impression of events (like receiving a bullet) and effects that should impress the user (like feeling an artificial heartbeat to add stress or suspense), but there is no consideration of obtaining feedback on the status of the user. Data about user physiology or motor responses, other than the inertial sensors (that cannot be considered as haptic feedback), could be used to assess the actual engagement of the user and even to prevent anomalous conditions, like heart conditions or cognitive conflicts. In some of the wide diversity of haptic system designs, implementation of a two-way information flow may be unfeasible. For instance, in the case of visual haptics systems, there is no actual way to receive such user’s haptic feedback.

Underlying the process of haptic data, whether its generation from virtual spaces into haptic effectors or its interpretation from actual haptic sensors, is the mapping of complex representations onto haptic data. Such representations and mappings are not general. Each application has a specific domain of representations, specific measured variables, and specific interpretations. It is, therefore, rather difficult to state something like a general theory of haptic science. Right now, most systems and solutions cannot be translated into different domains or even different kinds of users. The need for such a general theory of haptics may be underestimated in the current state of affairs, where many companies and research centers are pursuing divergent lines of research, but it may become more urgent as the systems increase their complexity and the population of users grows from limited ecosystems to the more general public.

Multimodality is key in the development of innovative interaction systems involving haptics. There is almost no purely haptic system in the market: it may only make sense for extremely limited subjects, such as blind–deaf persons, whose unique spatial sense is the sense of touch. Most current systems rely on redundant information that comes from the vision or the auditory sensors, or both. Such rich information needs also to be processed and/or generated in a coordinated manner. Conflicts between sensory information should be solved, in order to avoid cognitive distress in the user. For instance, the visual representation of empty spaces may conflict with haptic sensations of boundaries or, the other way around, visual limits such as walls should not be traversed freely, concerning haptic sensors. Hence, multimodality poses additional data-processing issues that can limit the improved experience of the user.

## 6. Challenges and Future Directions

New advances in artificial intelligence (AI) pose two-way challenges. On the one hand, there is the potential application of AI to the design and development of innovative haptic systems. Nowadays, haptic data are analyzed by means of statistical tools, including machine learning (ML) tools. Is it possible to enhance the analysis of haptic data by means of innovative AI architectures, such as large language models (LLMs)? LLMs are the main trend in current generative models for applications such as chatbots. Might it be possible to formulate haptic interactions in the framework of chatbot interactions? For instance, it might be possible to design innovative interaction systems capable of generating adequate spatial haptic stimuli triggered by gestural or environmental information, much like the current LLM-based chatbots generate text responses to text queries. Current multimodal foundational models, such as vision-to-language models (VLM) or language-to-vision models (LVM) that translate data between modalities, could be a certain inspiration for these innovative language-to-haptics models (LHM) and related developments.

Specifically, for blind–deaf people the main challenge is the construction of complex mental representations that can be expressed in language from the pure haptic experience. How to build and convey the representation of buildings, mountains, the sky, and other spatial concepts from pure tactile sensations? Related to this is the problem of communication. How to communicate complex ideas on the basis of pure haptic information?

On the other hand, the integration of haptic sensing into AI systems allows them to achieve a better grounding with reality. It is widely acknowledged that semantics are intimately related to sensory data, i.e., mental categories are often related to sensory stimuli. As an example, the terms “above” or “below” cannot be substantiated without some visual or haptic sensation that conveys the meaning of the spatial order implied by these terms. Without actual perception of objects placed one on top of the other, there is no actual meaning to “above” or “below”. In this regard, haptic sensing can provide a large variety of sensorial stimuli associated with otherwise obscure terms. The challenge in the not-so-far future will be to convey these meanings to general artificial intelligence (GAI) systems by the integration of haptic and other sensations. Many cognitive categories associated with pain and pleasure are grounded in haptic sensations that the GAI systems will need to master, in order to approximate human intelligence.

When considering children with developmental syndromes, such as ASD or ADHD, the big challenge is to tailor the systems to the specifics of the children. These developmental syndromes have a wide spectrum of actual symptoms and conditions. Hence, no single system with a rigid structure and definition of interactions and purposes can be of general benefit to any child diagnosed with these syndromes. Careful modeling of the child is required, and the field of continual learning should provide solutions to this conundrum. Haptics are a much more intimate interaction than vision-based systems, and they should prove valuable in such close modeling of the person. The integration of GAI systems and haptic solutions seems the most promising avenue of research to achieve a closely personalized assistant for children with developmental syndromes.

Haptics for enhanced immersion in gaming systems have attracted much attention, due to the large economical value of even small advances and improvements. It is to be expected that high-speed evolution of interaction devices for gaming will serve to lower the cost of individual units and to provide an exponential increase in their quality and offered features. In this context, it will be challenging to transfer gaming solutions into educative domains, specifically for the people at risk of exclusion. For instance, high-quality and high-resolution gaming suits, approximating an artificial skin enhanced with GAI, could create virtual educational environments for blind–deaf people.

## 7. Conclusions

This paper has visited the main topics related to haptic interfaces, with a specific emphasis on training and educational processes. Haptics have already demonstrated high value in some areas of training, especially medical and surgical training, where they enhance the virtual experience, improving the results of the training process. This paper indicates the usefulness of haptics-enriched systems in the realm of the education of persons at risk of exclusion. Some collectives, such as blind people, already benefit from this kind of system, although there is still much room for innovation. However, other collectives, such as children with development syndromes or other limitations, have received little attention. In the near future, it may be quite possible that solutions developed for gaming will be translatable into applications to these under-attended collectives. It is also rather attractive to work on the enhancements that new AI systems will contribute to educational processes based on multimodal virtual systems with high haptic components.

## Figures and Tables

**Figure 1 sensors-24-06946-f001:**
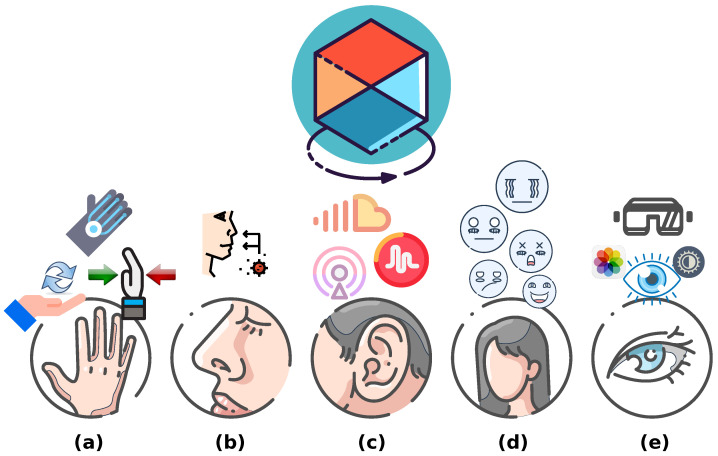
Environment–user virtual interactions: (**a**) Touching: Generating sensations such as pain, pressure, temperature, etc. [11,12,13]. (**b**) Smelling: Producing sensitivity to odors, aromas, flavors, etc. [14,15]. (**c**) Listening: Transmitting sounds, rhythms, tunes, etc. [16,17]. (**d**) Feelings: Eliciting emotions and comfort atmosphere [18]. (**e**) Vision: Generating accessible virtual environments [19,20,21].

**Figure 2 sensors-24-06946-f002:**
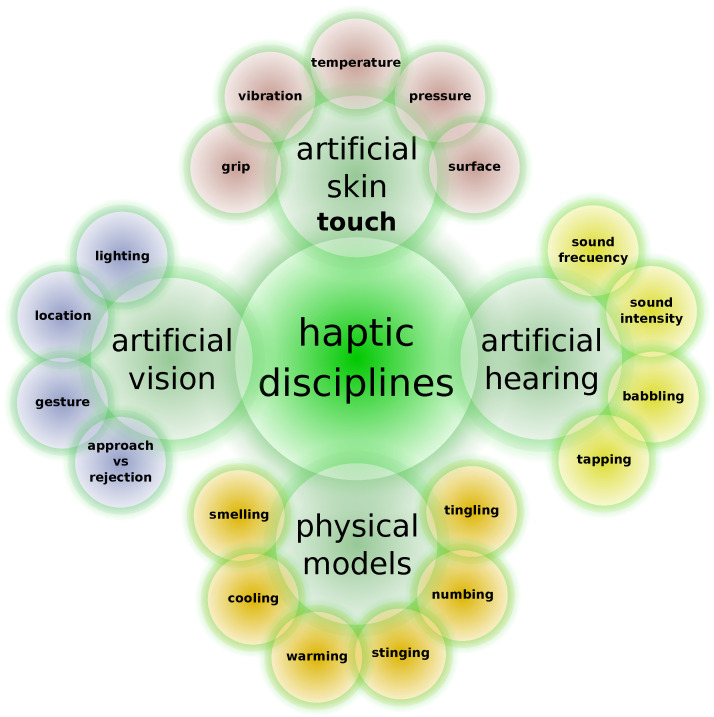
Haptic technologies and their relation with other technologies.

**Figure 3 sensors-24-06946-f003:**
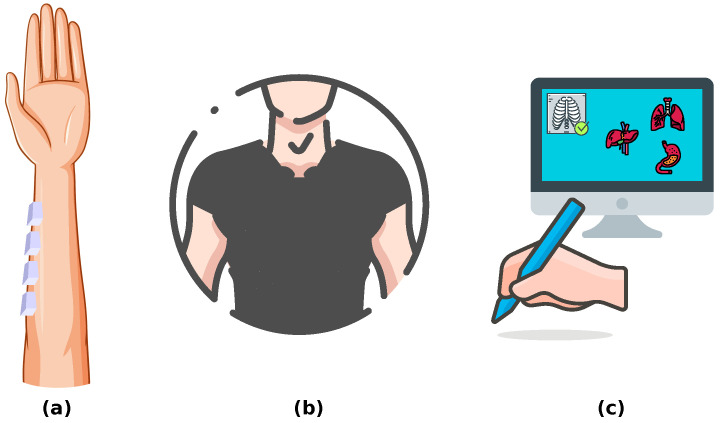
Multi-sensor/actuator-based medical developments. Wearable solutions to improve the perception processes: (**a**) TechARM: Electrode units to emit and record different stimuli [17]. (**b**) Teslasuit: Physical suit providing haptic feedback and capture motion and biometrics [85]. (**c**) Medical education enhanced by haptic devices [79].

**Figure 4 sensors-24-06946-f004:**
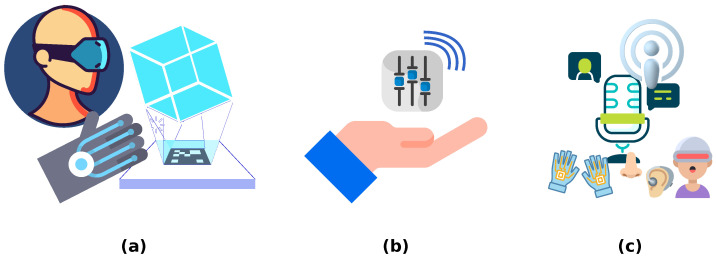
Interaction with the environment: (**a**) Meta—Inside Reality Labs: Bringing Touch to the Virtual World [31]. (**b**) Teenage Engineering Rumble: a bolt-on haptic subwoofer for the OP-Z [16]. (**c**) Hapticast: Haptic weekly podcast filled with gaming news [106].

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
