# Peer review of "A Narrative Review of Haptic Technologies and Their Value for Training, Rehabilitation, and the Education of Persons with Special Needs"

_sensors, 2024, doi:10.3390/s24216946_

Round 1

Reviewer 1 Report

Comments and Suggestions for Authors

The authors presented a literature summary on haptics, their classification and role in education and other major applications. The authors failed to discuss the key challenges and opportunities in the light of state-of-the-art. Please improve the technical quality in the respective areas.

Please improve the abstract, in the current form, the readers cannot get complete information about the purpose of the article. Cleary state the purpose and methodology/innovations.

Please improve the introductory section with some more additional literature. It is also necessary to cover the aspects of visual haptics and their role in education for low cognitive level persons.

In the industrial context, haptics is widely used in the training, guidance and maintenance tasks. The authors can closely look into it with the following keywords “Augmented reality-based guidance in product assembly and maintenance/repair perspective” “Haptics for industrial guidance”

Include more illustrations at section 2 and 3

Please expand the challenges with the existing technology with possible potential opportunities for the future.

Conclusions need to be expanded with future opportunities.

Author Response

Responses to reviewer #1 (Round 1)

The authors would like to thank the reviewer for her/his suggestions and comments, which have made it possible to significantly improve the article.

The authors presented a literature summary on haptics, their classification and role in education and other major applications. The authors failed to discuss the key challenges and opportunities in the light of state-of-the-art. Please improve the technical quality in the respective
areas.
Response: The manuscript has been fully rewritten. The changes are so big that we have not highlighted the paper changes in red, because everything (almost) would be red. We hope that the concerns of the reviewer have been addressed conveniently

Please improve the abstract, in the current form, the readers cannot get complete information about the purpose of the article.
Response: The abstract has been fully rewritten

Cleary state the purpose and methodology/innovations.
Response: We state clearly that the intent is to provide a narrative review. (Title and narrative)

Please improve the introductory section with some more additional literature. It is also necessary to cover the aspects of visual haptics and their role in education for low cognitive level persons.
Response: The introduction section has been rewritten completely, it remain short because we do not want to be redundant with other sections of the paper. We have added information about visual haptics.

In the industrial context, haptics is widely used in the training, guidance and maintenance tasks. The authors can closely look into it with the following keywords “Augmented reality-based guidance in product assembly and maintenance/repair perspective” “Haptics for industrial guidance”
Response: Thanks for the suggestion, we have added some paragraphs about the suggested topic but formal references are rather scarce and we have focused on pubmed references.

Include more illustrations at section 2 and 3
Response: We have changed the overall content of the manuscript, trying to convey a broad panorama of the haptics technologies and applications. Adding new figures has the inconvenient of dealing with copyright issues and permissions, which have not been granted in the cases we have asked for them.

Please expand the challenges with the existing technology with possible potential opportunities for the future.
Response: Thanks for the suggestion, we have included these discussions along the manuscript.

Conclusions need to be expanded with future opportunities.
Response: thanks for the suggestion, we have included these discussions along the manuscript.

Reviewer 2 Report

Comments and Suggestions for Authors

Dear authors 

review: 

Haptic Technology: A Valuable Paradigm for Reinforcing Learning Processes in Early Childhood Special Education

congratulation to your paper. 

Aims: to provide an overview of haptic technologies and how they present solutions that can be integrated into the aforementioned educational processes.

Aims: to motivate the study of educational tools that include haptic technologies for children with special needs. 

1. WELL STRUCTURED ARTICLE 

Some very important ideas: of paper: .... is to direct the attention of the scientific community so that  developments in the field of haptic technology are directed towards solutions that reinforce the aforementioned learning processes, incorporating methods and devices supported by Haptics that manage to improve the processes of communication and interaction with these children.  line 307 

excellent  and very inspirative :  Figure 1. Haptic Technologies 

Authors are experts who understand this problem: for example:  

. It is known that for children with special needs (such as ASD children), due 310 to their reduced communication skills, it is especially difficult to gain their attention and 311 maintain a traditional education and learning standards and that conventional educational processes are not useful. For them, educational technology developments based on haptic 313 solutions can offer new possibilities to keep these communication channels open. 

For better understanding implement some ideas: 

Weiss, E., Akimjaková, B., Paľa, G., & Biryukova, Y. N. (2023). Methods of Re-Education of Specific Learning Disorders. Journal of Education Culture and Society14(1), 185-197. https://doi.org/10.15503/jecs2023.1.185.197

concept of values: 

Kralik, R. (2023). The Influence of Family and School in Shaping the Values of Children and Young People in the Theory of Free Time and Pedagogy. Journal of Education Culture and Society14(1), 249-268. https://doi.org/10.15503/jecs2023.1.249.268 

Accept in present form

Author Response

Responses to reviewer #2 (Round 1)

The authors thank the reviewer for his suggestions and comments, which have opened up new challenges and new perspectives for the future work that we will continue to do.

Haptic Technology: A Valuable Paradigm for Reinforcing Learning Processes in Early Childhood Special Education

congratulation to your paper.

Response: Thank you very much for your opinion and appreciation.

Aims: to provide an overview of haptic technologies and how they present solutions that can be integrated into the aforementioned educational processes.

Aims: to motivate the study of educational tools that include haptic technologies for children with special needs.

  1. WELL STRUCTURED ARTICLE

Response: Thank you very much for this valuable opinion about our work.

Some very important ideas: of paper: .... is to direct the attention of the scientific community so that developments in the field of haptic technology are directed towards solutions that reinforce the aforementioned learning processes, incorporating methods and devices supported by Haptics that manage to improve the processes of communication and interaction with these children. line 307

Response: This is the crux of the matter, as was intended to be indicated in the initial aims of this contribution.

excellent and very inspirative : Figure 1. Haptic Technologies

Response: Thank you again for your positive feedback.

Authors are experts who understand this problem: for example:

. It is known that for children with special needs (such as ASD children), due 310 to their reduced communication skills, it is especially difficult to gain their attention and 311 maintain a traditional education and learning standards and that conventional educational processes are not useful. For them, educational technology developments based on haptic 313 solutions can offer new possibilities to keep these communication channels open.

Response: Thank you again for your positive feedback.

For better understanding implement some ideas:

Weiss, E., Akimjaková, B., Paľa, G., & Biryukova, Y. N. (2023). Methods of Re-Education of Specific Learning Disorders. Journal of Education Culture and Society, 14(1), 185-197. https://doi.org/10.15503/jecs2023.1.185.197

concept of values:

Kralik, R. (2023). The Influence of Family and School in Shaping the Values of Children and Young People in the Theory of Free Time and Pedagogy. Journal of Education Culture and Society, 14(1), 249-268. https://doi.org/10.15503/jecs2023.1.249.268

Response: Thank you for the references. Undoubtedly, these works will be taken into account in future studies and developments of the present work.

Accept in present form

Response: Thanks for the positive appraisal. We have rewritten the entire manuscript in order to answer the concerns of other reviewers. We expect to have answered also the comments of this reviewer.

Reviewer 3 Report

Comments and Suggestions for Authors

Proposing Haptic Technology to enhance Learning Processes in Education for People wiht Special Needs

Title, type

Types of "Review": (i) "State-of-the-art": Describe the current state of knowledge in a field. (ii)"Narrative review": Presents a chronological account of the research on a certain topic. (iii)  "Systematic review": Uses a rigorous methodology to synthesize and analyze data from randomized clinical trials. (iv) "Meta-analysis": Combines the statistical results of several similar studies.

What type of review do you propose in this article?

A scientific work of the "Review" type involves a critical and synthetic analysis of the existing literature on a certain subject or field. Unlike original research articles, which present new data obtained through experiments or studies, a "Review" does not primarily aim to generate new knowledge. Instead, he focuses on: Synthesizing information, Critically evaluating studies, Identifying trends and patterns, Identifying gaps in current knowledge, Formulating conclusions and recommendations.

So, I don't think that a Review article can have the word "proposing" in its title. This is the prerogative of some research works that, following some studies, propose a model that they verify. Only then can it be called a "proposal".

Abstract

An abstract of a scientific article is a concise synthesis of its content, presenting the main elements in a clear and concise way. The purpose of the abstract is to give the readers a quick picture of the topic addressed, the methodology used, the results obtained and the conclusions drawn.

Essential information to be covered in the abstract: Purpose of the study; Methodology; Results; Conclusions.

References: 90, almost good. Review articles usually have twice as many references. There may be the problem of the lack of articles in the field, being, however, a fairly new area of research.

Starting from the title, I would have seen the structure of the article differently:

1. Who are people with special needs?

2. How is the education of these people carried out?

3. What difficulties/problems were identified in the education process?

4. How to solve these identified problems using classical learning / education methods?

5. How classic methods can be replaced with Virtual Reality technologies, in general and haptic equipment, in particular.

6. What are the most suitable learning processes (with the fastest and quality results) when implementing haptic equipment?

After these activities, follows the study of haptic equipment that can be used for the stated purpose.

Otherwise, only chapter 4 would meet the requirements in the title regarding people with special needs...

The studied data are not structured, no indicators or characteristics are proposed that can be evaluated. The storytelling method is used instead of engineering methods for structuring information.

As a working method, you opted for a presentation of all the haptic equipment, with references in some scientific works which, then, in a very short analysis, you propose to be used in the education of people with special needs. It doesn't seem like a very rigorous approach to me.

What is the opinion of specialists in medicine / psychology / pedagogy about these equipments? Are they accepted by people with special needs? If they are refused? Have you assessed these risks?

Author Response

Responses to reviewer #3 (Round 1)

The authors would like to thank the reviewer for her/his suggestions and comments, which have made it possible to significantly improve the article.

Proposing Haptic Technology to enhance Learning Processes in Education for People wiht Special Needs

Title, type

Types of "Review": (i) "State-of-the-art": Describe the current state of knowledge in a field. (ii)"Narrative review": Presents a chronological account of the research on a certain topic. (iii) "Systematic review": Uses a rigorous methodology to synthesize and analyze data from randomized clinical trials. (iv) "Meta-analysis": Combines the statistical results of several similar studies.

What type of review do you propose in this article?

A scientific work of the "Review" type involves a critical and synthetic analysis of the existing literature on a certain subject or field. Unlike original research articles, which present new data obtained through experiments or studies, a "Review" does not primarily aim to generate new knowledge. Instead, he focuses on: Synthesizing information, Critically evaluating studies, Identifying trends and patterns, Identifying gaps in current knowledge, Formulating conclusions and recommendations.

So, I don't think that a Review article can have the word "proposing" in its title. This is the prerogative of some research works that, following some studies, propose a model that they verify. Only then can it be called a "proposal".

Response: Thanks for the educative comments. We have rewritten the manuscript as a “narrative review” following your advices.

Abstract

An abstract of a scientific article is a concise synthesis of its content, presenting the main elements in a clear and concise way. The purpose of the abstract is to give the readers a quick picture of the topic addressed, the methodology used, the results obtained and the conclusions drawn.

Essential information to be covered in the abstract: Purpose of the study; Methodology; Results; Conclusions.

References: 90, almost good. Review articles usually have twice as many references. There may be the problem of the lack of articles in the field, being, however, a fairly new area of research.

Response: Thank you for your helpful and educational comments. We have rewritten the abstract and found significant number of new references that we hope would approach the required quantity.

Starting from the title, I would have seen the structure of the article differently:

  1. Who are people with special needs?
  2. How is the education of these people carried out?
  3. What difficulties/problems were identified in the education process?
  4. How to solve these identified problems using classical learning / education methods?
  5. How classic methods can be replaced with Virtual Reality technologies, in general and haptic equipment, in particular.
  6. What are the most suitable learning processes (with the fastest and quality results) when implementing haptic equipment?

After these activities, follows the study of haptic equipment that can be used for the stated purpose.

Otherwise, only chapter 4 would meet the requirements in the title regarding people with special needs...

The studied data are not structured, no indicators or characteristics are proposed that can be evaluated. The storytelling method is used instead of engineering methods for structuring information.

As a working method, you opted for a presentation of all the haptic equipment, with references in some scientific works which, then, in a very short analysis, you propose to be used in the education of people with special needs. It doesn't seem like a very rigorous approach to me.

What is the opinion of specialists in medicine / psychology / pedagogy about these equipments? Are they accepted by people with special needs? If they are refused? Have you assessed these risks?

Response: Thank you for your comments. We have painstakingly rewritten the entire manuscript following your advice.

Round 2

Reviewer 1 Report

Comments and Suggestions for Authors

The manuscript is modified at most of the sections.

Discussions on the technological developments with regard to the disabilities should be highlighted.

The challenges involved in new developments should be clearly illustrated.

Please include a separate section to discuss about  “potential opportunities” /Future scope.

Please do verify the formatting corrections. (such as missing of full stop, comma, etc.,)

The literature is well covered, still some more additional suggestions for further inclusion.

Irigoyen, Eloy, Mikel Larrea, and Manuel Graña. "Haptic Technology: A Valuable Paradigm for Reinforcing Learning Processes in Early Childhood Special Education." (2024).

Wang, Qiang, Xiaofeng Xiang, and Bingcan Chen. "Food Protein Based Nanotechnology: from Delivery to Sensing Systems." Current Opinion in Food Science (2024): 101134.

Liang, Axin, et al. "Dynamic simulation and experimental studies of molecularly imprinted label-free sensor for determination of milk quality marker." Food Chemistry 449 (2024): 139238.

Sorgini, Francesca, et al. "Haptic-assistive technologies for audition and vision sensory disabilities." Disability and Rehabilitation: Assistive Technology 13.4 (2018): 394-421.

Author Response

Responses to reviewer #1 (Round 2-3)

The authors would like to thank the reviewer once again for his new suggestions and comments, which have made it possible to improve the article one more time.

The manuscript is modified at most of the sections.

Response: Thanks for the comment.

Discussions on the technological developments with regard to the disabilities should be highlighted.

Response: Thanks for the comments. We have changed the first two paragraphs in the new section “Challenges and future directions”, where new technological challenges in this area are raised.

The challenges involved in new developments should be clearly illustrated.

Response: Thank you for your comments.  We have modified the “challenges and future directions” section, emphasizing the challenges identified, as stated in the included sentences: “Artificial Intelligence (AI) new advances pose two way challenges. On the one hand, the potential application of AI for the design and development on innovative haptic systems.”; ”Specifically, for blind-deaf people the main challenge is the construction of complex mental representations that can be expressed in language from the pure haptic experience.”; “On the other hand, the integration of haptic sensing into AI systems allows them to achieve a better grounding with reality.  The challenge in not so far future will be to convey these meanings to general artificial intelligence (GAI) systems by the integration of haptic and other sensations.”; or “When considering children with developmental syndromes, such as ASD or ADHD, the big challenge is to tailor the systems to the specifics of the children.”.

Please include a separate section to discuss about “potential opportunities” /Future scope.

Response: Thank you again for your comments. We have added, reviewed and modified the new “challenges and future directions” section, as indicated in previous responses.

Please do verify the formatting corrections. (such as missing of full stop, comma, etc.,)

Response: Thanks for this comment. We have revised and proofread the entire manuscript to improve its readability once again.

The literature is well covered, still some more additional suggestions for further inclusion.

Irigoyen, Eloy, Mikel Larrea, and Manuel Graña. "Haptic Technology: A Valuable Paradigm for Reinforcing Learning Processes in Early Childhood Special Education." (2024).

Response: Thanks for the suggestion. However, this preprint corresponds to the early version of this paper. We do not think it is appropriate to self reference the work.

Wang, Qiang, Xiaofeng Xiang, and Bingcan Chen. "Food Protein Based Nanotechnology: from Delivery to Sensing Systems." Current Opinion in Food Science (2024): 101134.

Response: Thanks for the suggestion. However, we do not find solid connection of this reference to the current manuscript under review.

Liang, Axin, et al. "Dynamic simulation and experimental studies of molecularly imprinted label-free sensor for determination of milk quality marker." Food Chemistry 449 (2024): 139238.

Response: Thanks for this interesting suggestion. However, we also found no solid connection of this reference to the current manuscript under review.

Sorgini, Francesca, et al. "Haptic-assistive technologies for audition and vision sensory disabilities." Disability and Rehabilitation: Assistive Technology 13.4 (2018): 394-421.

Response: Thanks for the suggestion. This reference is very appropriate, so we have included it in the manuscript.

Reviewer 3 Report

Comments and Suggestions for Authors

Compared to the first version, the following have changed:

Compared to the first version, approx. 60 references and approx. 650 words in the text were additionally added.

The work is no longer specified as a "Review", it is noted as an "Article". I do not consider that this document is a research paper, it does not meet the recognized rules of a research paper. If it is considered to be a "Review" work, then the rules of this type of document presented on the mdpi.com website must be respected.

https://www.mdpi.com/about/article_types

Otherwise, the current work can be considered a monograph-type book chapter, which covers a lot of references, being up-to-date with the information, but which does not present Discussions, Conclusions or Future Development Directions.

Review

Reviews offer a comprehensive analysis of the existing literature within a field of study, identifying current gaps or problems. They should be critical and constructive and provide recommendations for future research. No new, unpublished data should be presented. The structure can include an Abstract, Keywords, Introduction, Relevant Sections, Discussion, Conclusions, and Future Directions.

Other sources:

https://esrj.edu.af/esrj/review, https://authorservices.taylorandfrancis.com/publishing-your-research/writing-your-paper/how-to-write-review-article/ , https://www.ncbi.nlm.nih.gov/pmc/articles/PMC3715443/

Author Response

Response to reviewer 3 (Round 2)

The authors would like to thank the reviewer once again for his new suggestions and comments, which have made it possible to improve the article one more time.

Compared to the first version, the following have changed:

Compared to the first version, approx. 60 references and approx. 650 words in the text were additionally added.

The work is no longer specified as a "Review", it is noted as an "Article". I do not consider that this document is a research paper, it does not meet the recognized rules of a research paper. If it is considered to be a "Review" work, then the rules of this type of document presented on the mdpi.com website must be respected.

https://www.mdpi.com/about/article_types

Response: Thanks for the recommendations. The paper has been formatted as a “Narrative review” and we have followed the rules to the best of our knowledge. The specification as an “Article” was an error during the submission process that we will correct in the resubmission.

Otherwise, the current work can be considered a monograph-type book chapter, which covers a lot of references, being up-to-date with the information, but which does not present Discussions, Conclusions or Future Development Directions.

Response: Thanks for the comments and observations. We have included the required sections. However, please note that the description that you reproduce below states “The structure can include…” Definitively it is not written: “The structure must include…” Nevertheless, we have striven to comply with the reviewer’s comments.

Review

Reviews offer a comprehensive analysis of the existing literature within a field of study, identifying current gaps or problems. They should be critical and constructive and provide recommendations for future research. No new, unpublished data should be presented. The structure can include an Abstract, Keywords, Introduction, Relevant Sections, Discussion, Conclusions, and Future Directions.

Other sources:

https://esrj.edu.af/esrj/review, https://authorservices.taylorandfrancis.com/publishing-your-research/writing-your-paper/how-to-write-review-article/ , htps://www.ncbi.nlm.nih.gov/pmc/articles/PMC3715443

Response: Thanks for the comments. We have included a new and extended section “Challenges and future directions”, where new technological challenges in this area are raised.
